# Antimicrobial Resistance Patterns and Clonal Distribution of *E*. *coli*, *Enterobacter* spp. and *Acinetobacter* spp. Strains Isolated from Two Hospital Wastewater Plants

**DOI:** 10.3390/antibiotics11050601

**Published:** 2022-04-29

**Authors:** Miguel Galarde-López, Maria Elena Velazquez-Meza, Miriam Bobadilla-del-Valle, Patricia Cornejo-Juárez, Berta Alicia Carrillo-Quiroz, Alfredo Ponce-de-León, Alejandro Sassoé-González, Pedro Saturno-Hernández, Celia Mercedes Alpuche-Aranda

**Affiliations:** 1Centro de Investigación Sobre Enfermedades Infecciosas, Instituto Nacional de Salud Pública, Morelos 62100, Mexico; miguel.galarde@insp.edu.mx (M.G.-L.); berta.carrillo@insp.mx (B.A.C.-Q.); 2Laboratorio Nacional de Máxima Seguridad para el Estudio de Tuberculosis y Enfermedades Emergentes, Instituto Nacional de Ciencias Médicas y Nutrición “Salvador Zubirán”, Mexico City 14080, Mexico; mbv99@hotmail.com (M.B.-d.-V.); alf.poncedeleon@gmail.com (A.P.-d.-L.); 3Departamento de Infectología, Instituto Nacional de Cancerología, Mexico City 14080, Mexico; patcornejo@yahoo.com; 4Unidad de Inteligencia Epidemiológica, Hospital Regional de Alta Especialidad de Ixtapaluca, Ixtapaluca 56530, Mexico; sassoe777@hotmail.com; 5Centro de Investigación en Evaluación de Encuestas, Instituto Nacional de Salud Pública, Morelos 62100, Mexico; pedro.saturno@insp.mx

**Keywords:** *Enterobacteriaceae*, wastewater, antibiotic resistance, WWTP, public health

## Abstract

The objective of this study was to determine the presence and persistence of antimicrobial-resistant enterobacteria and their clonal distribution in hospital wastewater. A descriptive cross-sectional study was carried out in wastewater from two Mexico City tertiary level hospitals. In February and March of 2020, eight wastewater samples were collected and 26 isolates of enterobacteria were recovered, 19 (73.1%) isolates were identified as *E. coli*, 5 (19.2%) as *Acinetobacter* spp. and 2 (7.7%) as *Enterobacter* spp. Antimicrobial susceptibility profiles were performed using the VITEK 2^®^ automated system and bacterial identification was performed by the Matrix-Assisted Laser Desorption/Ionization-Time of Flight mass spectrometry (MALDI-TOF MS^®^). ESBL genes were detected by polymerase chain reaction (PCR) and clonal distributions of isolates were determined by pulsed-field gel electrophoresis (PFGE). *E. coli* susceptibility to different classes of antimicrobials was analyzed and resistance was mainly detected as ESBLs and fluoroquinolones. One *E. coli* strain was resistant to doripenem, ertapenem, imipenem and meropenem. The analysis by PCR showed the presence of specific β-lactamases resistance genes (*bla_KPC_*, *bla_CTX-M_*). The PFGE separated the *E. coli* isolates into 19 different patterns (A–R). PFGE results of *Acinetobacter* spp. showed the presence of a majority clone A. Surveillance of antimicrobial resistance through hospital wastewater is an important tool for early detection of clonal clusters of clinically important bacteria with potential for dissemination.

## 1. Introduction

Antimicrobial resistance (AMR) is a growing public health problem worldwide due to the presence of multidrug-resistant (MDR) bacterial pathogens in healthcare settings [1]. The World Health Assembly issued the “Global Action Plan on Antimicrobial Resistance” to address the AMR problem [2], urging countries to strengthen surveillance to direct targeted research and interventions appropriated for each country or even institutions. In 2017, the World Health Organization published a list of pathogens of priority attention, among which are carbapenem-resistant and extended-spectrum β-lactamase *Enterobacteriaceae* [3].

Antimicrobial-resistant enterobacteria producers of extended-spectrum β-lactamase (ESBL) are the most frequently isolated microorganisms in healthcare units; these are highlighted by the presence of several enzymes that can hydrolyze penicillin and inhibit penicillin and broad-spectrum beta-lactam antibiotics, such as third- and fourth-generation cephalosporins and monobactams. Different variants of ESBL (e.g., TEM, TLA, SHV, CTX-M) have been described with worldwide dissemination, including Mexico [4,5]. Despite the availability of a few antibiotics treatments for ESLB-producing enterobacteria, there is emerging resistance, which limits patients’ therapeutic options and is associated with high morbidity and mortality [6].

The presence of antimicrobial-resistant enterobacteria is frequently described in hospital settings; however, their presence in other locations or settings, such as community, agricultural, and environmental settings, demonstrates the ability of enterobacteria to spread beyond healthcare settings [7,8]. Some studies carried out in hospitals in Mexico report data on *E. coli* resistant to beta-lactams with alarming values (50%), as well as an increase in resistance to carbapenems in *Enterobacter* spp. isolates [9,10]. In this regard, hospital wastewater plays an essential role in the spread of antimicrobial-resistant bacteria, including *Enterobacteriaceae*, because wastewater carries and receives both bacteria and resistance genes and antibiotic concentrations from healthcare institutions. Therefore, investigating the presence of antimicrobial-resistant bacteria in hospital and/or community wastewater through an epidemiological surveillance system makes it possible to monitor, prevent, alert and control the spread of antibiotic resistant bacteria both in the hospital environment and in the environment [11,12,13].

Some hospitals have wastewater treatment plants (WWTPs), and different authors have described that WWTP treatments significantly reduce the abundance of bacteria. The main objective of WWTPs is to remove organic matter and most environmental microorganisms [14]. Unfortunately, WWTPs do not eliminate all antibiotic-resistant bacteria and their resistance genes, so the persistence of pathogenic bacterial after treatment is feasible. This poses a paradigm for public health epidemiological surveillance based on hospital wastewater as an alternative approach and rapid assessment due to the excretion of AMR and partially metabolized antimicrobials in patients’ urine and feces through the sewer [15,16]. 

In this paper we describe the antibiotic resistant and clonal profiles of *Escherichia coli, Acinetobacter* spp. and *Enterobacter* spp. in raw and treated wastewater from two hospitals.

## 2. Results

### 2.1. Enterobacteriaceae Isolates

Eight wastewater samples were collected from the influent and affluent of the WWTPs of two third-level attention hospitals in Mexico. A total of 26 enterobacteria were presumptively identified using selective chromo-agar and subsequently confirmed by MALDI-TOF MS ^®^ and VITEK 2^®^; 19 (73.1%) isolates were identified as *E. coli*, 5 (19.2%) as *Acinetobacter* spp. and 2 (7.7%) as *Enterobacter* spp. Half of the isolates were isolated from raw wastewater and the other half were from treated wastewater. In Hospital A, eight *Escherichia coli* and one Entererobacter cloacae were isolated from the raw wastewater and three *E. coli*, one *E. bugandensis* and five *Acinetobacter* spp. (*A. haemolyticus* and *A. modestus*) were isolated from the treated wastewater. At Hospital B, four *E. coli* were isolated from the raw wastewater, four *E. coli* were isolated from treated wastewater, and no isolates of other enterobacteria were detected in this hospital.

### 2.2. Antimicrobial Susceptibility

Antimicrobial susceptibility results detected by VITEK 2^®^ showed that 11 strains were resistant to ampicillin-sulbactam (MIC_90_ of >32 mg/L); five strains were resistant to piperacillin-tazobactam (MIC_90_ of >128 mg/L); and eight strains were resistant to cefoxitin, ceftazidime, ceftriaxone and cefepime (second-, third- and fourth-generation cephalosporin) (MIC_90_ of >64 mg/L). In addition, one *E. coli* strain was resistant to doripenem, ertapenem, imipenem and meropenem (MIC_90_ of ≤0.12, ≤0.5, ≤0.25 and ≤0.25 mg/L, respectively); eleven strains were resistant to ciprofloxacin (MIC_90_ of >4 mg/L) and two strains were resistant to gentamycin (MIC_90_ of <1 mg/L). Both *Enterobacter* spp. isolates were resistant to cefoxitin (MIC_90_ of >64 mg/L). All *Acinetobacter* spp. isolates were susceptible to tygecyline and colistin, and three isolates were amikacyn-resistant (MIC_90_ of >64 mg/L) (Table 1). 

### 2.3. Detection of β-Lactamases Resistance Genes

The molecular analysis by PCR showed the presence of specific β-lactamases resistance genes (*bla_KPC_*, *bla_CTX-M_*); it was observed that one *E. coli* strains carried *bla_KPC_*, and another harbored *bla_CTX-M_*. Two isolates were also observed to harbor more than one gene (Table 1).

### 2.4. Molecular Typing by Pulsed-Field Gel Electrophoresis (PFGE)

The PFGE separated the *E. coli* isolates into 19 different patterns (A–R) without subtypes; the patterns A–K were from Hospital A and the patterns L–R were from Hospital B. The patterns A–H and L–Ñ were present in *E. coli* isolates from raw wastewater, whereas the patterns I–K and O–R were isolates from treated wastewater. The PFGE separated the *Enterobacter* spp. into two patterns (S and T): one pattern in raw wastewater and one pattern in treated wastewater from Hospital A (Figure 1). 

PFGE results of *Acinetobacter* spp. isolates showed the presence of a majority clone A (n = 4) and an unrelated pattern (B); these clones were detected in strains collected in treated wastewater from Hospital A (Figure 2).

The PFGE patterns analysis showed similarity percentages from 73% to 38% in *E. coli* and *Enterobacter* spp. and similarity percentages that were from 100% to 28% in *Acinetobacter* spp.

When PFGE patterns were compared with antimicrobial susceptibility profiles, it was observed that *E. coli* strains with patterns F, G, I, K, L, M, O and P showed resistance to a higher number of antimicrobials tested, whereas strains with patterns A, C, E, J, N, Ñ and R were sensitive to all antimicrobials tested. The patterns T and S of *Enterobater* spp. had the same resistance profile. On the other hand, in *Acinetobacter* spp. clone A, resistance to amikacin was observed in three of the four strains tested, whereas clone B was sensitive to all antimicrobials tested.

## 3. Discussion

The high frequency of *E. coli* strains in raw and treated hospital wastewater samples, compared to other bacteria, would be expected in some extent, mainly because it is one of the bacterial species that are part of the microbiota of the digestive tract of healthy and sick humans and is widely distributed in the environment. However, its dissemination and spread in hospital wastewater highlights the risk of contamination of aquatic environments or bodies of water by possible pathogenic strains of *E. coli* resistant to antimicrobials [17]. Although the objective of this study was not to determine whether *E. coli* strains isolated from hospital wastewater were pathogenic or not, it was revealed that they were the main bacterial species showing multidrug resistance (n = 7) and there was a presence of extended-spectrum beta-lactamases in three *E. coli* strains tested. The *E. coli* resistance patterns were relatively similar between the two hospitals studied (Table 1).

Resistance of *E. coli* to different classes of antimicrobials, such as beta-lactams and fluoroquinolones, mainly amoxicillin and ciprofloxacin, was evident in raw and treated wastewater samples from both hospitals (A and B), which may be due to the selection pressure of antimicrobial-resistant strains in the clinical hospital environment and their persistence through the treatment processes in the wastewater treatment plants. Several studies in different countries have reported a similarly high rate of resistance among *E*. *coli* strains to beta-lactams of clinical and environmental origin [18,19]. This is associated with the antimicrobial resistance mechanism generated by beta-lactamase enzymes, encoded by resistance genes and carried on plasmids, which can be exchanged between different bacterial species by horizontal transmission of genes (HTG) and mobile genetic elements (MGE) [20].

The two isolates of the *Enterobacter* spp. isolated in this study were resistant to second-generation cephalosporins (Table 1). Barraud et al. [21] reported one strain of *Enterobacter cloacae* resistant to quinolones, beta lactams, aminoglycosides and tetracycline obtained from hospital residual water. A carbapenem-resistant strain of *E. coli* was detected in the raw wastewater at Hospital B. Although resistance to carbapenems due to the presence of carbapenemase enzymes has been widely described in *Enterobacteriaceae*, mainly in genera such as *K. pneumoniae*, it has also been reported in *E. coli* and *Enterobacter* spp. isolated from sewage. This is of concern because *Enterobacteriaceae* are responsible for infectious outbreaks and carbapenem antibiotics are generally used as a last resort in the treatment of diseases caused by bacteria resistant to other classes of antibiotics [7]. *Enterobacter* spp. has been claimed to be a bacterial genus that serves as a reservoir of antimicrobial resistance genes through HTG and MGE [8].

Only isolates of the *Acinetobacter* spp. were evaluated against colistin, and they were found to be sensitive (Table 1). Overall, the highest proportion of isolates was sensitive to aminoglycosides and fully sensitive to tigecycline and this differs from the findings of Wang et al. [22] and Zhang et al. [23], who reported one and ten strains of *Acinetobacter* spp., respectively, with a resistance profile to cephalosporin and carbapenems. In this work we reported three strains of *A. haemolyticus* with resistance to amikacin (MIC range 32–64 mg/L). Similar results were reported by Zong et al. [23], Bardhan et al. [19] and Kovacic et al. [24], who detected amikacin-resistant *Acinetobacter* spp. in hospital wastewater.

Although in this study, eight primers were used for the detection of β-lactamases resistance genes including a mixture of the *AmpC* gene, we only detected the presence of *bla_KPC_* and *bla_CTX-M_* genes in *E. coli* strains. Compared to *K. pneumoniae* isolates, *bla_KPC_* producing *E. coli* strains remain less frequent [25]. However, *bla_CTX-M_* producing *E. coli* were more frequently reported. The first report on the *bla_CTX_*_-_*_M_* gene was in 1989, with CTX-M being the most prevalent ESBL in the world [26]. We found that two strain of *E. coli* carried *bla_KPC_* and *bla_CTX-M_*, and this differs from the findings of Kutilova et al. [27], who reported a high prevalence of *bla_CTX-M_* (>95%) in strains of *E. coli* isolated from raw hospital wastewater. Xu et al. [25] reported similar results to ours in samples of river water where wastewater is discharged, reporting three strains of *E. coli* carrying *bla_KPC_* and *bla_CTX-M_*. This is in agreement with the results reported by Zumaya et al. regarding the excess use of cephalosporins and carbapenems in the medical prescription of the hospitals included in this study [28]. 

Molecular typing by PFGE was performed to investigate the clonal relationship between the *E. coli* strains, *Enterobacter* spp. and *Acinetobacter* spp. (Figure 1 and Figure 2). In our study, we detected a remarkable diversity of patterns in *E. coli* strains and *Enterobacter* spp. (n = 21); this heterogenicity of patterns has been observed in other studies, such as those published by Galvin et al. [29] and Zhao et al. [30], who found 18 PFGE patterns among the 23 isolates analyzed and 7 PFGE patters in 9 strains studied isolated from hospital WWTPs. However, it is remarkably interesting that antimicrobial resistance profiles are highly similar in the different clones. Thus, these data suggest that these antimicrobial resistance genes are highly selected in *Enterobacteriaceaes* in the hospital environment. Another of our important findings was that the strains of *A. haemolyticus* that carried the same PFGE pattern were found in treated wastewater, showing its ability to persist in its passage through the WWTP. The clonality of *Acinetobacter* spp. has been document in other studies. In studies carried out in Croatia and India, PFGE revealed related patterns among *A. baumannii* strains [19,31]. In another works in China, PFGE analysis showed pattern heterogeneity, this being different from our results [23,32].

## 4. Materials and Methods

### 4.1. Study Sites and Samples Collection 

The study was conducted in two third-level hospitals in the Mexico City Metropolitan Zone in February and March 2020. One is a national referral hospital (Hospital A) with 119 beds, the other is a regional hospital (Hospital B) with 246 beds located in the State of Mexico, which provides specialized care to people from several states in the center of the country. Single samples of raw wastewater and treated wastewater were the primary sampling sources. Wastewater released from both hospitals flows directly to a wastewater treatment plant, and then to the city sewer.

Hospital wastewater samples were collected from the influent and effluent of each wastewater treatment plant (WWTP). A total of eight wastewater samples were collected over three weeks, two raw and two treated wastewater samples each from Hospital A and B. Wastewater simple samples were obtained by the “grap” technique in sterile 1 L containers [14], and transported at <4 °C to the Research Center for Infectious Disease at the National Institute of Public Health in Cuernavaca, Morelos within two hours of collection for processing.

### 4.2. Microbial Culturing and Identification

Microbial culturing of the *Enterobacteriaceae* family was performed by taking an aliquot of 100 mL of raw wastewater and 200 mL of treated wastewater in 50 mL conical bottom propylene tubes, which were centrifuged for 20 min at 5000 revolutions per minute (rpm). After decanting the supernatant, the pellet was dissolved in 10 mL of phosphate buffer solution (PBS) to homogenized by shaking.

After homogenization and prior to plating, the samples were diluted 1:1000 with PBS, the following agar plates were plated by single streaking: Hi-Crome™ ECC (HiMedia Laboratories, Mumbai; India), Koser Citrate Medium^®^ and Hi-Crome™ Acinetobacter Agar Base (HiMedia Laboratories, Mumbai; India) inoculating 100 μL of solution. The plates were labeled with the type of wastewater, sampling location, date and sample number. Plates were incubated at 37 °C, and results were read after 18 hours. Different presumptive colonies of each bacterial species were selected based on the color patterns described by the manufacturer on chromogenic agars. They were then transferred to MacConkey agar at 37 °C for 18 h for purification and identification. The isolates were identified by mass spectrometry, using the Microflex MALDI-TOF MS^®^ (Bruker Daltonics, Bremen, Germany) equipment.

### 4.3. Antimicrobial Susceptibility Testing

Antimicrobial susceptibility profiling of strains was carried out by a VITEK 2^®^ automated system (BioMe’rieux, Marcy l’Etoile, France), according to the break point established by the CLSI 2021 guidelines [33]. Fourteen antimicrobials were tested: ampicillin-sulbactam, piperacillin-tazobactam, cefoxitin, ceftazidime, ceftriaxone, cefepime, doripenem, ertapenem, imipenem, meropenem, amikacin, gentamicin, ciprofloxacin and tigecycline.

### 4.4. Detection of β-Lactamases Genes 

The presence of β-lactamases genes in *Enterobacteriaceae* isolates was confirmed by PCR. The assay included primers to detected the following β-lactamases genes: *bla_CTX-M_, bla_SHV_, bla_AMPc_, bla_GES-2_, bla_KPC_, bla_OXA-48-like_, bla_NDM-1_* and *bla_IMP_* described previously [34,35,36,37,38,39,40,41]. DNA templates were used by PCR analysis and the products were visualized in 1% agarose gel.

### 4.5. Molecular Typing of PFGE and Computer Fingerprint Analysis 

After digestion with *Xba* I and *Apa* I endonucleases, the DNA was separated using a CHEF-DR II system (BioRad, Birmingham, UK) [42,43]. The *Salmonella* serotype Braenderup strain (H9812) was included in each PFGE gel as an internal control. Computer analysis of PFGE profiles was done using the Gel Compar II software, v.6.6.11 (Applied Maths, Inc.; Sint-Martens-Latem, Belgium) after visual inspection using the criteria of Tenover [44]. The Dice coefficients were calculated and were then transformed into an agglomerative cluster by the unweighted pair group method with arithmetic average (UPGMA).

## 5. Conclusions

Our results highlight the potential threat of extended-spectrum beta-lactamase resistance gene transmission in raw and treated hospital wastewater, as well as the risk of dissemination to environmental reservoirs such as rivers and other water bodies. The wide heterogeneity of PFGE pathways shown by *E. coli* strains may favor the selection of several genotypes, mainly those that showed resistance to the antibiotics tested. The clonality shown in *A. haemolyticus* strains isolated from treated wastewater suggests that this bacterium has a genetic potential to survive hospital WWTP treatment systems. Surveillance of antimicrobial resistance through hospital wastewater is an important tool for early detection of clonal clusters of clinically important bacteria with potential for dissemination.

## Figures and Tables

**Figure 1 antibiotics-11-00601-f001:**
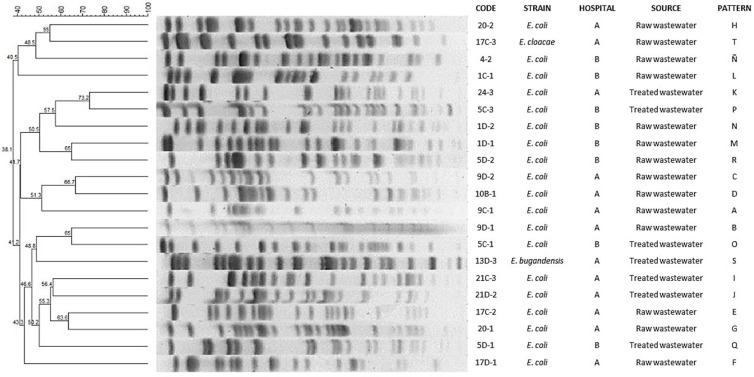
Dendrogram based on pulsed-field gel electrophoresis (PFGE) patterns after digestion with enzyme *Xba* I of *E. coli* and *Enterobacter* spp. isolates isolated from hospital wastewater.

**Figure 2 antibiotics-11-00601-f002:**
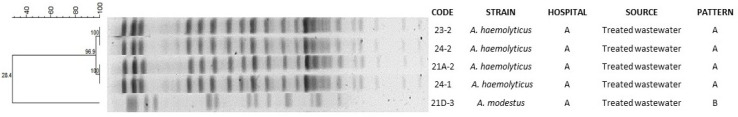
Dendrogram based on pulsed-field gel electrophoresis (PFGE) patterns after digestion with enzyme *Apa* I of *Acinetobacter* spp. isolates isolated from hospital wastewater.

**Table 1 antibiotics-11-00601-t001:** Phenotyping properties of *E. coli*, *Acinetobacter* spp. and *Enterobacter* spp. isolated from the hospital wastewater treatment plants (WTTP).

Hospital	Wastewater	Code	Isolated	Genes	ESBL	AMP	TZP	FOX	CAZ	CTX	CEF	DOR	ERT	IPM	MEM	AMK	GEN	CIP	TIG	COL
**A**	Raw	9C-1	*E. coli*		-															
		9D-1	*E. coli*		-															
		9D-2	*E. coli*		-															
		10B-1	*E. coli*		-															
		17C-2	*E. coli*		-															
		17D-1	*E. coli*		-															
		20-1	*E. coli*	*bla_KPC_*																
		20-2	*E. coli*	*bla_CTX-M_*	+															
		17C-3	*E. cloacae*																	
	Treated	21C-3	*E. coli*	*bla_KPC,_bla_CTX-M_*	+															
		21D-2	*E. coli*		-															
		24-3	*E. coli*		-															
		21A-2	*A. haemolyticus*																	
		21D-3	*A. modestus*																	
		23-2	*A. haemolyticus*																	
		24-1	*A. haemolyticus*																	
		24-2	*A. haemolyticus*																	
		13D-3	*E. bugandensis*																	
**B**	Raw	1C-1	*E. coli*		-															
		1D-1	*E. coli*		-															
		1D-2	*E. coli*		-															
		4-2	*E. coli*		-															
	Treated	5C-1	*E. coli*		+															
		5C-3	*E. coli*	*bla_KPC,_bla_CTX-M_*	-															
		5D-1	*E. coli*		-															
		5D-2	*E. coli*		-															

Green: Susceptible; Red: Resistant; Yellow: Intermediate; White: Not performed/Not determined. Ampicillin-sulbactam (AMP), piperacillin-tazobactam (TZP), cefoxitin (FOX), ceftazidime (CAZ), ceftriaxone (CTX), cefepime (CEF), doripenem (DOR), ertapenem (ERT), imipenem (IPM), meropenem (MEM), amikacin (AMK), gentamycin (GEN), ciprofloxacin (CIP), tigecycline (TIG) and colistin (COL).

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
