# Peer review of "Antimicrobial Resistance Patterns and Clonal Distribution of E. coli, Enterobacter spp. and Acinetobacter spp. Strains Isolated from Two Hospital Wastewater Plants"

_antibiotics, 2022, doi:10.3390/antibiotics11050601_

Round 1

Reviewer 1 Report

  • The term BLEES needs to be changed into ESBL since it is the international term of it.
  • In the results part, the authors need to give an intro instead of directly mentioning that they got 26 isolates. The intros should inform the sampling location, brief isolation method and the process how can the authors at the end got 26 isolates.
  • Regarding the process how the authors got 26 isolates, it needs to be shown in the supplementary file.
  • In line 115-116 the authors stated that sequencing analyses are still underway. This sentence is not supposed to be written in the submitted manuscript. It would be better if the author either wait for the sequencing results to come out and add them in this manuscript or just do not mention it at all.
  • The authors provide the table of resistance and susceptibility of the isolated strains with the detected resistant genes. In the material and method part, the authors used many primers pairs to detect the genes but the detected genes were only 2 among 14 genes. However, the authors did not discuss it thoroughly about why only these 2 genes are detected.
  • In material and method, despite the authors mentioned that they used Salmonella DNA as a control for PFGE experiment, they did not include it in both figure 1 and 2. Therefore, I suggested to include the control in the figures.
  • There are some grammatical errors found throughout the manuscript. Please correct and proofread.

Author Response

Thank you for your valuable time and comments.

Reviewer 2 Report

 The presentation of results in the manuscript is excellent, and this is a good example of the contamination of environmental water bodies from wastewater from hospitals via extended-spectrum beta-lactamase resistance gene transmission in raw and treated hospital wastewater.

All standard methods were used for the experiments and data collection. I found the document interesting for the readers and follow the scope of the journal Antibiotics.

I would recommend the article could be published in Antibiotics, after a minor revision. There are technical errors, I hope the editor will take care of them.

The authors need to address the below-mentioned queries.

  1. Line 41: For “In 2015, the World Health Assembly”: Any recent global action plan addressing the issue of antimicrobial resistance by the World Health Assembly.
  2. The author could include the location of the two hospitals in the title.
  3. Although the samples were taken in February and March of 2020, however, the author took two years to communicate the results, is the analysis of the results required a longer time?
  1. In section: 1. Enterobacteriaceae Isolates and 2.2. Antimicrobial Susceptibility. A tabular form of results could be a better presentation.
  2. Line 240: Change 37°C to 37°C (need space); for Line 260: space missing−20◦C; please check throughout the manuscript.

6. Use either hours or h for time throughout the manuscript.

7. The author needs to mention “table/figure number and entries” in the discussion section while discussing the results for better understanding.

8. The author could include the following relevant references.

(a) Poirel L, Madec JY, Lupo A, Schink AK, Kieffer N, Nordmann P, Schwarz S. Antimicrobial Resistance in Escherichia coli. Microbiol Spectr. 2018 Jul;6(4). doi: 10.1128/microbiolspec.ARBA-0026-2017. PMID: 30003866.

(b) Kazemnia, Ali et al. “Antibiotic resistance pattern of different Escherichia coli phylogenetic groups isolated from human urinary tract infection and avian colibacillosis.” Iranian biomedical journal vol. 18,4 (2014): 219-24. doi:10.6091/ibj.1394.2014

(c) Paitan Y. Current Trends in Antimicrobial Resistance of Escherichia coli. Curr Top Microbiol Immunol. 2018;416:181-211. doi: 10.1007/82_2018_110. PMID: 30088148.

(d) American medical centers with a diagnosis of pneumonia: analysis of results from the SENTRY Antimicrobial Surveillance Program (1997). SENTRY Latin America Study Group. Diagn Microbiol Infect Dis. 1998 Dec;32(4):289-301. doi: 10.1016/s0732-8893(98)00124-2. PMID: 9934546.

Author Response

Thank you for your valuable time and comments

Reviewer 3 Report

The objective of this study was to determine the presence and persistence of antimicrobial-resistant enterobacteria and their clonal distribution in raw and treated wastewater from two hospitals. the proposed work is very interesting and the paper is well written. However, 

  1. The number of collected samples is very low and collected in very short time intervals.
  2. you did not state in the abstract, what is bacterial prevalence?
  3. in section 2.2, you need to describe the % of resistance for each antibiotic
  4. you need to run make correlation analysis between phenotypic and genotypic resistance
  5. you need to add statistical analysis as a separate section in materials and methods and also run stats on your samples

Author Response

(The authors gave the same response as above.)

Round 2

Reviewer 1 Report

All of the concerns have been addressed.

Author Response

Thank you for your review of the manuscript, your comments have enriched our work.

Reviewer 3 Report

I still have a big concern about the number of the samples as this did not allow you to run any stats on your results

Author Response

Thank you for your review of the manuscript, your comments have enriched our work.
In relation to your concern, we would like to let you know that with this work we intended to perform a first descriptive evaluation of the importance of hospital wastewater in Mexico, which would allow us to generate a first evidence of the national context. With this and considering your comments, we will now be able to propose future longitudinal studies that show statistically significant differences in the sampling sites.